# Prenatal Caffeine Exposure Is Linked to Elevated Sugar Intake and BMI, Altered Reward Sensitivity, and Aberrant Insular Thickness in Adolescents: An ABCD Investigation

**DOI:** 10.3390/nu14214643

**Published:** 2022-11-03

**Authors:** Khushbu Agarwal, Peter Manza, Hugo A. Tejeda, Amber B. Courville, Nora D. Volkow, Paule V. Joseph

**Affiliations:** 1Section of Sensory Science and Metabolism, National Institute on Alcohol Abuse and Alcoholism, National Institutes of Health, Bethesda, MD 20892, USA; 2National Institute of Nursing Research, National Institutes of Health, Bethesda, MD 20892, USA; 3Laboratory of Neuroimaging, National Institute on Alcohol Abuse and Alcoholism, National Institutes of Health, Bethesda, MD 20892, USA; 4Unit on Neuromodulation and Synaptic Integration, National Institute of Mental Health, National Institutes of Health, Bethesda, MD 20892, USA; 5National Institute of Diabetes and Digestive and Kidney Diseases, National Institutes of Health, Bethesda, MD 20892, USA

**Keywords:** prenatal caffeine exposure, total sugar intake, body mass index, reward sensitivity, taste processing

## Abstract

Prenatal caffeine exposure (PCE) has been positively associated with elevated body mass index (BMI) in children. Why this association occurs is unclear, but it is possible that PCE alters the in utero development of brain structures associated with food preference, leading to more total sugar intake (TSI, grams) later in childhood. To test this hypothesis, we investigated if PCE (daily/weekly/<weekly vs. no exposure) and elevated BMI are associated with increased TSI, neural activation during large reward anticipation (monetary incentive delay task—functional MRI) and structural changes (thickness, mm) in taste processing regions of children (*n* = 5534; 9–11 years) from the large-scale Adolescent Brain Cognitive Development (ABCD) study. Linear mixed-effect models, after covariate adjustments, identified a positive association (*p* < 0.05, all |βs| > 0.01) of excessive PCE (vs. no exposure) with elevated BMI (daily/weekly/daily limit; consistent in boys and girls), increased TSI (daily) and insular thickness (daily/weekly), as well as low middle frontal cortex (MFC) activation (daily). Our sub-analysis revealed an association of daily/weekly PCE (vs. no exposure) with increased gram sugar intake from soft drinks. We also identified a positive relationship of excessive PCE with elevated TSI and increased insular thickness (a key gustatory region), while in a Sobel test, reward sensitivity (reduced brain reactivity to reward anticipation in MFC; tracks reward outcomes) mediated (Test statistic = 2.23; *p* = 0.02) the PCE-linked BMI changes in adolescents. Our findings suggest that excessive PCE might be detrimental to frontal lobe development and altered reward sensitivity to food, thereby increasing risk for elevated TSI and obesity. Our results support recommendations to limit caffeine intake during pregnancy.

## 1. Background

Proper nutrition during pregnancy is crucial for both the mother and the child’s health. Brain development in children is impacted by prenatal intake of compounds, such as folate, mercury, iron, omega fatty acids, as well as caffeine [1,2,3,4,5]. Caffeine is a naturally occurring central nervous system stimulant that is found in tea and coffee, but it is also added to food and beverages [6]. As such, it is consumed widely, including by approximately 75 percent of pregnant women in the United States [7,8,9,10]. However, as caffeine metabolism in the liver is reduced during pregnancy, its unmodified form passes readily through the membranes of the placenta and infant blood brain barrier [7,8,9,10]. Therefore, an intake threshold of no more than 200 mg/day caffeine has been suggested, as greater consumption adversely affects the development of the fetus and/or postnatal growth [7,8,9,10]. Consequences of prenatal caffeine exposure (PCE) on children include maladaptive behavior and cognition [11,12], developmental delays, abnormal neuro-motor activity [13], excess body weight [14] and a heightened risk for obesity [15,16]. 

The prevalence of obesity in children and adolescents (5–19 years) has significantly increased worldwide over the past two decades, with approximately 124 million and 213 million current cases of obesity and overweight, respectively, occurring in this age group globally [17]. The association between excessive total sugar intake (TSI) and body mass index (BMI) in children and adults has been inconsistent across the literature [18], with some reports of no association [19], but with others showing a negative [20] or a positive association [21,22]. Boys reportedly consume higher amounts of sugar [23,24] and increased total relative energy intake from sugar than girls [24]. Alterations in reward sensitivity are related to increased dietary sugar intake in children [25]. However, heightened sugar intake could also be driven by alterations in sweet taste perception and sensitivity [26]. The processing of taste is regulated in the brain by the insular cortex [27]. Sweet taste perception is peripherally recognized by the tongue’s sweet taste receptors and these signals are transmitted through the brain stem and thalamus to the primary gustatory cortex, which comprises the frontal operculum and anterior insula. The anterior insula and the associated gustatory cortex respond to the taste and physical properties of food and may also respond to its reward value [28]. Prefrontal cortical structures (PFCs; mediodorsal and ventrolateral in rodents, orbitofrontal in primates) are involved in relaying the taste information to the reward system and the feeding center [29].

While it is known that maternal caffeine intake is associated with increased obesity risk in children, the mechanisms for this association remain unclear. It is plausible that PCE alters reward sensitivity and changes insular structure, thereby increasing a child’s preference for foods rich in sugar [30]. Here, we sought to test this hypothesis using the large Adolescent Brain Cognitive Development (ABCD) dataset. By doing so, we established relationships of PCE with (i) BMI status; (ii) TSI; (iii) functional brain activity in different regions of interest (ROIs) to large reward anticipation during the monetary incentive delay (MID) task; and (iv) thickness changes in taste-processing regions. We also investigated if TSI, altered reward sensitivity during anticipation of large rewards or altered insular thickness in these adolescents serve as potential mediators underlying the relationship of PCE with BMI.

## 2. Methodology

### 2.1. Data Source 

The Adolescent Brain Cognitive Development ^SM^ (ABCD) Study is an ongoing longitudinal study focused on brain development and child health in the United States, with the inclusion of over 11,000 children 9–10 years old through adolescence [31]. To maximize the recruitment of a diverse demographic and socioeconomic U.S. population, the recruitment of the study cohort involved 21 sites, following a rigorous recruitment strategy, including epidemiologically informed school-based sampling [32]. Data collection was performed once parents’ provided consent and children’s assent. Yearly multimodal assessments were conducted, including a wide range of measures, environmental, behavioral, physical health, and neurocognitive as well as structural and functional MRI scans every two years. The complete details of the ABCD Study with complete description on the study design, recruitment strategy, sample selection procedure and complete battery of assessments are provided elsewhere [33]. The exclusion criteria adopted at the time of recruitment for the ABCD study were diagnosis of any underlying conditions, including (i) schizophrenia, (ii) moderate to severe autism spectrum disorder, (iii) moderate to severe intellectual disability, (iv) major neurological disorders or (v) a substance use disorder. Further, participants were also excluded if they had signs of non-correctable vision, hearing, or sensorimotor impairments, had a gestational age less than 28 weeks with a birth weight < 1.2 kg, any parent-reported birth complications that required more than a month of hospitalization, any history of traumatic brain injury or any standard MRI contraindications (such as metal implants, claustrophobia, orthodonture). The study procedures were approved by the Institutional Review Boards of the participating study site and by the centralized Institutional Review Board of the ABCD study. Each participant, including the caregivers and the children, approved and provided written consent and verbal assent, which was approved by a central institutional review board for the ABCD study (https://abcdstudy.org/sites/abcd-sites.html, accessed on 9 October 2022).

### 2.2. Data Selection

The dataset was obtained from the ABCD 2.0.1 data release, which contains 11,875 children at ages 9–11. In the large ABCD study, children were excluded from enrollment if they lacked English proficiency or had severe sensory, intellectual, medical, or existing neurological conditions. The first acquisition of the dataset, including the interviews, questionnaires (both parents and children) and MRI scans (only children), were completed in about 6–7 h during their first visit in 1- or 2-day sessions. As the block food questionnaire was first introduced during the two-year follow-up session, we considered this acquisition as the first time point measurement for this variable. The analysis in the present study was conducted on 5534 adolescents after excluding 6344 children with mothers with confirmed alcohol consumption or use of any other illicit drugs, with no TSI data, and those with no data on caffeine use during pregnancy. 

### 2.3. Prenatal Caffeine Exposure (PCE)

Prenatal exposure to caffeine (coffee and tea) was assessed with the Developmental History Questionnaire [34]. The categories of PCE (no/daily/weekly/less-than-weekly exposure) used in our analysis was based on the question ‘Did you/biological mother have any caffeine during pregnancy (from conception until delivery)?’ (0 = No, 1 = Yes—at least once a day; 2 = Yes—less than once a day but more than once a week; 3 = Yes—less than once a week). The daily caffeine intake question was also used to assess the dose differences on the outcome variables. Daily cup intake of up to 2 cups (recommended safe dose; *n* = 1069) and 3+ cups (excessive dose; *n* = 138) was used as sub-categories to determine dose effects. Mothers reported the number of weeks after conception they learned of their pregnancy in the questionnaire, with the average number of weeks reported being 6.7 ± 6.6.

### 2.4. Block Kids Food Screener Questionnaire (BKFS) (TSI (g))

The BKFS instrument was used to collect adolescents’ TSI filled by the parents/caregiver. This is a semi-quantitative questionnaire validated in children that ascertains the previous week’s frequency (from ‘none’ to ‘every day’) and consumption amount of 77 common food items (with three to four categories related to food type) [35]. It contains foods identified by National Health and Nutrition Examination Survey (NHANES) 2002–2006 commonly consumed by children. For each child, completed food-frequency questionnaires (FFQs) were analyzed by Nutrition Quest (Berkley, CA, USA). Daily fruit and vegetable intake was estimated in cup-equivalent servings. Vegetable servings exclude potatoes and legumes, and fruit servings include 100% fruit juice. Nutrition Quest also calculated the total sugars (grams). We used total sugars in foods and juices (grams) as estimates of sugar intake (i.e., TSI), as well as different food items listed as potential sugar source, including (note: spelled as found in data dictionary) fruitjuice, softdrinks, applebananaorange, applesaucecannedfruit, anyotherfruit, ketchupsalsa, icecream, candynbars, cookiedonutcake, cerealwgsweet, milkchocolate, breakfastproteinbars and wholewheatbread, in our analysis. 

### 2.5. Body Mass Index (BMI)

At enrollment BMI measurements of these children were used. BMI was calculated from measured height and weight: 703 × weight (lbs.)/height (in)^2^ (https://www.cdc.gov/nccdphp/dnpao/growthcharts/training/bmiage/page5_2.html, accessed on 1 July 2022).

Data for two children with impossible BMI values of 0 kg/m^2^ were excluded from the analysis.

### 2.6. Monetary Incentive Delay Task Functional MRI

A version of the MID task was used to measure brain activation during anticipation and receipt of three conditions [31]: reward (USD 0.20 or USD 5), loss (−USD 0.20 or −USD 5) or no incentive (USD 0). We focused on two primary contrasts: (1) anticipation of reward vs. no incentive and (2) anticipation of large reward vs. no incentive. Brain activity was examined in the following regions of interest (ROIs): bilateral prefrontal cortex; PFC (middle frontal cortex; MFC (rostral, caudal), anterior cingulate cortex; ACC (dorsal, caudal), orbitofrontal cortex (lateral, medial)); ventral striatum; VS (including the nucleus accumbens; NAc); amygdala; insula; and thalamus. Our selection of the PFC regions was based on prior literature reporting on the role of MFC, ACC and OFC in reward anticipation [36,37,38]. 

#### Structural Imaging

All the imaging data were preprocessed by the ABCD data team using standardized processing pipelines [39]. The present study used post-processed structural (i.e., cortical thickness) data based on the Desikan–Killiany brain registration atlas [40]. Cortical thickness (bilateral total) of taste-processing regions [41] including the insula, OFC and ACC was used in the current analysis.

### 2.7. Covariates 

Parents reported children’s age (months), sex (male or female) and race/ethnicity (White, Black, Latino/Hispanic or Other), highest household education (up to HS Diploma/GED, some college, bachelor, postgraduate), household income (<USD 50 K, ≥USD 50 and <10 K, ≥USD 10 K) and physical activity (number of physically active days for at least 60 min/day in the past 7 days) at enrollment.

### 2.8. Statistical Analyses

#### 2.8.1. Demographics

A chi-square test was conducted to determine differences in categorical variables (sex, race/ethnicity, household income, highest household educational level), while F-statistics were performed to assess the association between numeric variables (age, physical activity) and PCE groups, respectively. Categorical variables were dummy coded. 

#### 2.8.2. Association Analysis

Linear mixed-effect (LME) models were built to first assess the relationship between PCE and BMI in the dataset controlling for the confounding effects of age, sex, child race/ethnicity, highest household education, household income and physical activity. Further separate LME models were built to investigate the relationships between PCE and the following outcome measures: (i) TSI, (ii) functional brain activity during reward anticipation; and (iii) cortical thickness of taste-processing regions hypothesized to be potential mediators of BMI elevation in excessive PCE children. In all these models PCE categories, age, sex, child race/ethnicity, highest household education and household income were used as fixed effects while the study recruitment site was used as a random effect. A sub-analysis was conducted to understand the differential TSI intake and BMI in children with no exposure to caffeine in utero; exposed to recommended dose of 200 mg per day or up to two cups per day prenatal caffeine [42]; and with those above this limit of PCE and no exposure to caffeine in utero. Further exploration was undertaken to investigate the association of gram intake from different food items in BKFS, including (note: spelled as found in data dictionary) fruitjuice, softdrinks, applebananaorange, applesauce, anyotherfruits, ketchup salsa, icecream, candybars, cookiedonuts, cerealwithsweet, milkchocolate, breakfastprotein and wholewheatbread, with PCE categories. For this investigation we used the food items as outcome variables in LME model with PCE categories and other covariates as fixed effects and study recruitment site as random effect. We built similar LME models to examine the relationship of BMI with TSI, brain activation and thickness across various ROIs. For all models investigating associations with BMI, physical activity was incorporated as an additional covariate. When significant, post hoc tests were performed for pairwise group comparisons (two-sided and Bonferroni-corrected).

#### 2.8.3. Mediation Analysis

We also conducted mediation analysis using the Sobel test to identify if TSI, activation to large reward anticipation or thickness changes in ROIs were potential mediators of the association between PCE (daily vs. weekly/less than weekly/no exposure) and BMI. Additionally, we looked at whether activation to large reward anticipation or thickness changes in ROIs mediate the effect of PCE on TSI in our study cohort. 

#### 2.8.4. Sex Effects

Prior studies have reported sex differences in BMI [43] and sugar intake [23,24]; therefore, a sensitivity analysis was conducted to test a possible moderation effect of sex on the association of PCE and TSI with BMI. We also conducted a stratified analysis for boys and girls. LME was built with age, child race/ethnicity, highest household education and household income as covariates and study recruitment site as a random effect. All statistical analyses were performed using SPSS 28.0 (IBM Corp., New York, NY, USA).

## 3. Results

### 3.1. Demographic Characteristics of Children in PCE Groups

In the population analyzed, *n* = 2222 (40.15%) children had no in utero caffeine exposure; *n* = 1304 (23.56%) were exposed for at least once a day (0.25–10 cups/day); *n* = 877 (15.85%) had more than once per week (0.5–24 cups/week) but less than once per day and *n* = 1131 (20.44%) had less than once per week (1–32 cups/month). Racial/ethnicity, household income and household education differences were seen between the PCE categories. No significant age and sex differences were seen across children in different PCE categories (Table 1).

### 3.2. Association of PCE with BMI

PCE was positively associated with BMI (F_3,5076_ = 4.7; *p* = 0.002) after adjusting for all covariates. Daily and weekly PCE groups were associated with higher BMI while the no-exposure group was not ((β = 0.45; 95% CI (0.19, 0.71); *p* < 0.001), (β = 0.28; 95% CI (0.01, 0.55); *p* = 0.03); Table 2, Figure 1A). The mean difference in BMI between daily and no-exposure groups was higher than between the weekly and no-exposure group (Table 2). Significantly higher BMI levels were seen in both above and within daily limit PCE groups when compared to no-exposure children ((β = 0.93; 95% CI (0.29, 1.57); *p* = 0.004), (β = 0.45; 95% CI (0.17, 0.73); *p* = 0.001)), respectively (Table 2; Figure 2A). Post hoc pairwise comparisons on Bonferroni correction revealed a similar trend of mean difference in BMI between above daily limit vs. no exposure and within daily limit vs. no-exposure groups (Table 3). 

### 3.3. Association of PCE with TSI

Analyses of the relationship between varying levels of PCE with TSI, after adjusting for all covariates, showed a significant association of TSI with PCE in a dose-dependent manner (F_3,5509_ = 3.18; *p* = 0.02). Children with daily PCE were associated with higher TSI compared to children with no exposure (β = 3.51; 95% CI (1.21, 5.80); *p* = 0.003; Figure 1B). In contrast, less-than-weekly or weekly exposure groups did not show significant differences in TSI compared to the no-exposure group (Table 2; Figure 1B). The above safe daily consumption limit PCE subgroup (≥3 cups) had greater TSI (vs. no-exposure group β = 9.78, 95% CI (3.83, 15.74), *p* = 0.001; Table 2; Figure 2B). Interestingly, post hoc pairwise comparisons on Bonferroni correction revealed greater mean differences in TSI in PCE children with above daily consumption PCE children (*p* = 0.004) than those exposed within a daily limit (*p* = 0.03; Table 3). Interestingly, our exploration analysis revealed a significant association between gram intake of soft drinks (F_3,4948_ = 14.9; *p* < 0.001) and PCE. Group differences in soft drink intake were observed after adjusting for all covariates and correcting for multiple comparisons, such that daily and weekly PCE was associated with greater soft drink intake (vs. no exposure, β = 30.3, 95% CI (21.45, 39.29), *p* < 0.001; β = 12.77, 95% CI (3.47, 22.07), *p* = 0.007) (Appendix A). 

### 3.4. Association of PCE with Brain Activation during Anticipation of Large Reward and Structural Change in Taste-Processing Regions

We then examined the association of PCE with brain activation changes during reward anticipation and thickness changes in taste-processing regions, after adjusting for covariates. A significant negative association was seen between PCE levels with rostral MFC activation in anticipation of large reward (F_3,3650_ = 3.1; *p* = 0.02). Daily PCE (vs. no exposure) was associated with lower MFC activation (β = −0.027; 95% CI (−0.05, −0.002); *p* = 0.02) (Table 2; Figure 1C). However, post hoc pairwise comparisons after Bonferroni correction revealed no significant difference between daily PCE vs. no-exposure groups, while we noticed significant difference in MFC activation between the daily vs. less-than-weekly PCE-exposed children (Table 3). We did not observe significant activation differences for other ROIs. There were no significant differences in MFC activation between daily exposure sub-groups (Figure 2C). 

PCE was positively associated with insular thickness (F_3,5067_ = 2.79; *p* = 0.03); however, the association with PCE did not persist after adjusting for race/ethnicity. Daily and weekly PCE exposure (vs. no exposure) was associated with greater insular thickness ((β = 0.01; 95% CI (0.0008, 0.019); *p* = 0.03); (β = 0.01; 95% CI (0.0006, 0.02); *p* = 0.03)), respectively (Table 2; Figure 1D). However, post hoc pairwise comparisons did not reach significance after Bonferroni correction. There were no significant differences in insular thickness between daily exposure sub-groups (Figure 2D). 

### 3.5. Association of BMI with TSI, Brain Activation during Anticipation of Large Reward and Structural Change in Taste-Processing Regions

BMI was significantly associated with TSI (β = −0.003; 95% CI (−0.006, −0.0005); *p* = 0.02) after adjusting for the effects of all covariates. We observed a significant positive relationship between BMI and rostral MFC activation on large reward anticipation (β = −0.42; 95% CI (−0.790, −0.054); *p* = 0.02). However, BMI was not found to be associated with insular thickness in the study cohort. 

### 3.6. Exploratory Mediation Effects

We did not identify a significant indirect effect of TSI on the relationship between PCE and BMI in the study cohort by Sobel test. However, we observed an indirect significant effect of rostral MFC activation on large reward anticipation in the relationship between PCE and BMI (Sobel Test statistic = 2.23; *p* = 0.02). Furthermore, we tested whether lower rostral MFC activation or insular thickness mediates the effect of PCE on TSI in children. A marginally significant indirect effect attributed to rostral MFC activation was seen with the relationship between PCE and TSI (Sobel Test statistic = 1.89; *p* = 0.07). However, this result did not persist after covariate adjustment.

### 3.7. Sex Effects

Although we did not observe a significant interaction of sex with PCE (F_3,5076_ = 1.21, *p* = 0.30) and TSI (F_1,5083_ = 0.72, *p* = 0.39) on BMI as an outcome, a stratified analysis showed a consistent significant relationship of PCE and BMI in both girls (F_3,2370_ = 2.8, *p* = 0.04) and boys (F_3,2685_ = 3.1, *p* = 0.02) after covariate adjustment. While in girls, weekly PCE (vs. no exposure) was significantly associated with elevated BMI (β = 0.44; 95% CI (0.04, 0.84); *p* = 0.03), in boys, we noted that daily PCE (vs. no exposure) was associated with elevation in their BMI (β = 0.53; 95% CI (0.17, 0.88); *p* = 0.003). Notably, the relationship of PCE and BMI did not survive multiple comparison on Bonferroni correction in girls, but the results persisted in boys. Further, a significant relationship of TSI with BMI was seen in girls (F_1,2379_ = 5.8, *p* = 0.01) but not in boys (F_1,2696_ = 0.73, *p* = 0.39).

## 4. Discussion

We observed a significant positive relationship of excessive PCE with BMI, TSI and MFC activation on large anticipation, as measured by fMRI, as well as insular thickness changes in the ABCD population, controlling for the effects of known confounders. Specifically, our findings revealed a positive association between daily and weekly PCE children and elevated BMI levels. The elevated BMI seen in daily PCE children in our study corroborates reports from a recent study by Zhang et al. using the same dataset [11]. Furthermore, our results identified consistent evidence for elevated BMI in girls and boys with excessive PCE (weekly/daily vs. no exposure). 

Although the association with TSI in the above daily limit PCE subgroup of children was numerically stronger than the within daily limit children vs. no-exposure group, the association with BMI revealed a similar trend. A systematic review showed evidence of elevated BMI in children exposed to high levels of caffeine in utero (between 50 mg and <150 mg/day) and for increased risk for obesity at intakes ≥ 300 mg/day [16]. However, we are not aware of prior work demonstrating increased dietary sugar intake in children with excessive PCE. A study by Li et al. hypothesized the potential effect of added sugar in coffee of mothers to the observance of increased obesity risk in children with PCE. Though their results revealed an association between PCE and obesity risk in children up to 87%, even after controlling for factors related to maternal obesity (pre-pregnancy BMI) and metabolic disorders (preexisting diabetes and gestational diabetes), the causal effect of added sugar in caffeinated drinks in risk for child obesity could not be demonstrated [15]. Although we did identify a relationship between BMI and TSI, especially in girls in our study cohort, the absence of a potential mediating effect of TSI on the existing relationship between PCE and BMI was inconsistent with our hypothesis. 

One mechanistic hypothesis for elevated TSI and lower PFC activation on large reward anticipation in children with daily PCE (vs. no exposure) could be a negative impact of maternal caffeine on inhibitory control [44], which elevates their drive to consume obesogenic foods. Individual differences in sensitivity to rewards are stated as an important predictor of reactivity to one’s food environment [45]. Using the Behavioral Inhibition/Behavioral Activation Scales’ (BIS/BAS) as a measure of reward sensitivity, a significant positive association of reward sensitivity with fast food and sweet drink consumption frequency was previously seen in children (aged 5.5–12 years) [46]. As for PCE, it was demonstrated that excessive maternal caffeine intake during pregnancy has a long-lasting impact on children’s behavioral and neurocognitive development [11,47]. The many adverse effects of PCE on offspring cognition and overall brain development were studied in preclinical settings, revealing that self-administration of caffeine at chronic doses (2–3 cups/day coffee/day in humans) in rats altered the development of hippocampus and prefrontal cortex of offspring, impacting 24 h memory retention in the novel object recognition task and spatial learning in the radial arm maze [48]. Of note, caffeine consumption in children influences their decision-making and risk-taking behaviors [49]. Our results identified lower PFC (MFC) activation during anticipation of large reward as a mediator between the excessive PCE and BMI elevation in children. Although this study cannot confirm a causal relationship, based on the current data and prior reports in the scientific literature, we propose that daily/excessive PCE negatively impacts the child’s frontal lobe development, leading to lower inhibitory control and altered reward sensitivity in childhood, as revealed by lower PFC activation to large reward anticipation, thus, elevating their risk for obesity. However, changes in reward sensitivity cannot directly cause obesity. Therefore, further investigations considering different types of reward, such as screen time exposure or chemical substance usage, are needed to clarify the inner structure of reward sensitivity and its role as a moderator in elevated BMI in children with excessive PCE. 

Furthermore, our findings demonstrated a positive association between insular thickness and daily and weekly PCE, possibly reflecting altered sweet taste processing as another potential mechanism for elevated sugar intake in chronic PCE children. Cortical thickness measurement is used as a biomarker of divergent adolescent development, as thickness changes relate to neurocognitive traits (e.g., impulsivity, intelligence, ADHD and depression onset) [50,51,52,53]. Heightened cortical thickness in the insula is implicated in numerous addictive disorders in youths, such as substance use [54] and internet gaming disorders [55,56]. Insular activity appears to maintain a craving state, enhancing the drive to consume substances or play games. Moreover, the insula is involved in processing of taste signals [57] and bottom-up detection of salience events [58,59]. Thus, it is plausible that greater cortical thickness in the high-PCE groups (daily and weekly) reflects greater insular activation to gustatory signals, whereas their reduced middle PFC reactivity interferes with inhibitory control, unbalancing and enhancing reward sensitivity to sweets with impaired self-regulation, leading to elevated intake of sugary food and, thus, greater BMI. The results obtained in the ABCD children did not identify insular thickness alteration as a mediator of BMI elevation in children with excessive PCE and warrants further investigation in another large dataset.

We noted, through our exploratory analysis, an association between elevation in soft drink intake with daily PCE (vs. no exposure) in the study cohort. This finding aligns with previous reports that indicate sweetened beverages as the leading source of added sugar across all age groups in the United States [23,60,61,62] and in adolescents, with soft drinks (i.e., sodas, pops, and colas) as the key contributor to the sweetened beverage category. Therefore, it is possible that the elevated TSI seen in children with daily PCE is largely attributed to their higher soft drink intake. However, we did not identify soft drink intake as a potential mediator of the relationship between PCE and elevated BMI in these children. 

Taken together, our findings support the hypothesis that adolescents prenatally exposed to excessive doses of caffeine show elevated BMI and TSI, as well as impairments in reward processing and inhibitory control. While we identified altered reward sensitivity as an indirect mediator of the relationship between excessive PCE and elevated BMI of the ABCD children, TSI did not emerge as significant in the mediation analysis. We will conduct future investigations to identify the reinforcers that contribute to BMI elevation in children. Our work has clinical implications and helps to set the stage for additional investigation in this direction. Specifically, it is crucial to disentangle how the alteration in reward value anticipation mediates the elevation in BMI of children exposed to high in utero caffeine levels. This investigation will be essential to fully understand the mechanisms linking PCE and increased BMI in this crucial developmental period and may be key to changing eating and lifestyle habits, promoting lifelong health benefits. 

### Strengths and Limitations

A key strength of this study is the involvement of adolescents aged 9 and 11 years, which enabled us to understand the early effects of PCE on TSI and the risk for obesity. It is important to follow-up these children to see if the elevation in TSI persists in the daily PCE group and if they continue to show elevated BMI. However, at present, we could not assess this hypothesis using the ABCD dataset due to a lack of longitudinal data for these children. Even so, our results set the stage for future work for a deeper understanding of the later consequence of excessive sugar consumption on obesity development and other health-related problems when these children transition to adulthood. Another underlying strength is the use of a large, diverse, national dataset that has statistical power to detect relatively small effects across multiple sociodemographic groups. A small effect size was observed for our main findings regarding the relationship of PCE (daily vs. no exposure) with BMI (β = 0.45; η_p_^2^ = 0.001) and TSI (β = 3.5; η_p_^2^ = 0.0001). Similar effect sizes have been reported across several other studies using the ABCD dataset to evaluate the effects of PCE on behavioral outcomes in children (β = 1 in the whole sample and β = 1.3 in boys (daily vs. no exposure) [11]; β = 1.23 [63]; β = 0.1–2.0 [64]), which is argued to pertain to unequal representation of children and families in the ABCD dataset. 

The study is limited by the lack of maternal measurements, including the amount of sugar intake during pregnancy, as well as precise amounts of daily caffeine intake. Future studies should also consider maternal dietary behaviors, such as food consumption, as well as BMI and other anthropometric measures, as these are other factors that impact the BMI of children. Our results for MFC activation and insular thickness association with PCE did not survive on multiple comparison using Bonferroni correction and require further investigation using other large datasets. Furthermore, future investigations will benefit from longitudinal measurement of TSI and BMI of these children to derive relationships and possible variations from one age to another.

## 5. Conclusions

The findings reveal elevated BMI and TSI in children exposed in utero to daily and above-safe-limit levels of caffeine. Daily PCE (*vs*. no exposure) was also associated with lower MFC activation to large reward anticipation and increased insular thickening. Altered MFC activation in large reward anticipation showed as an indirect mediator between high BMI and excessive PCE in the children of the ABCD study. These findings demonstrate that chronic caffeine intake during pregnancy could be one factor in elevating the risk of obesity in children. The current recommendations on caffeine consumption for pregnant women need to be carefully assessed and an awareness program should be initiated to avoid adverse consequences in child development. 

## Figures and Tables

**Figure 1 nutrients-14-04643-f001:**
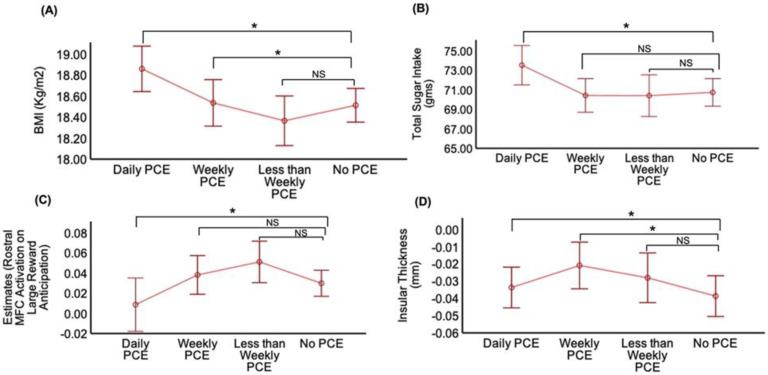
Line plots illustrating the association outcomes of BMI (**A**), total sugar intake (**B**), rostral middle frontal cortex activation on large reward anticipation during MID task-fMRI (**C**), insular thickness (**D**) in various prenatal caffeine exposure (PCE) (vs. no PCE) groups after controlling for the confounding effects of age, sex, child race/ethnicity, highest household education, household income and physical activity across different on linear mixed effect models. Here, * denotes statistical significance with *p* < 0.05 on Bonferroni correction, while NS denotes no statistical significance.

**Figure 2 nutrients-14-04643-f002:**
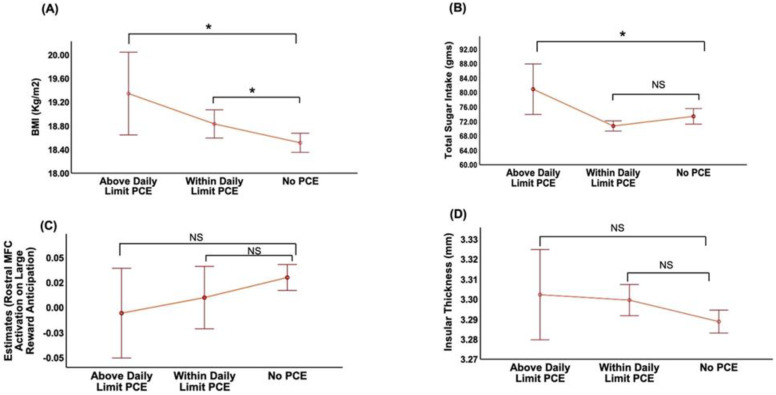
Line plots illustrating the association outcomes of BMI (**A**), total sugar intake (**B**), rostral middle frontal cortex activation on large reward anticipation during MID task-fMRI (**C**), insular thickness (**D**) in various daily prenatal caffeine exposure (PCE) (vs. no PCE) groups after controlling for the confounding effects of age, sex, child race/ethnicity, highest household education, household income and physical activity across different linear mixed-effect models. Here, * denotes statistical significance with *p* < 0.05 on Bonferroni correction, while NS denotes no statistical significance.

**Table 1 nutrients-14-04643-t001:** Demographics of children with different PCEs.

	No (*n* = 2222)	Less than weekly (*n* = 877)	Weekly (*n* = 1131)	Daily (*n* = 1304)	Chi-Square/F Statistics	*p* Value
Age (months) ^a^	119.35 (7.53)	119.71 (7.28)	119.73 (7.35)	119.42 (7.24)	0.937	0.422
Female ^b^	1031 (46.4)	537 (61.2)	410 (36.3)	614 (47.1)	0.394	0.94
Race/Ethnicity ^b^						
Black	333 (14.99)	87 (9.92)	75 (6.63)	110 (8.44)	123.509	<0.001
White	1129 (50.81)	755 (86.09)	527 (46.59)	783 (60.05)		
Hispanic	504 (22.68)	168 (19.16)	161 (14.24)	246 (18.87)		
Other	253 (11.39)	121 (13.79)	114 (10.08)	164 (12.58)		
NA	3 (0.14)	-	-	1 (0.08)		
Household Income ^c^	2.16 (0.83)	2.27 (0.78)	2.17 (0.83)	2.26 (0.78)	29.336	<0.001
Highest Household Education ^d^	2.88 (1.04)	3.01 (0.96)	2.84 (1.04)	3.01 (0.97)	44.746	<0.001
Physical Activity ^e^	3.63 (2.31)	3.5 (2.31)	3.64 (2.26)	3.66 (2.26)	34.716	0.03

Note—Prenatal caffeine exposure: No, No exposure; Daily, at least once a day; Weekly, less than once a day but more than once a week; Less than weekly, less than once a week. *p* Value of F-tests are reported for age. *p* value for Chi-square is reported for categorical variables sex, race/ethnicity, household income, highest household education, physical activity. Nonsignificant comparisons are not listed. ^a^ mean (SD). ^b^ N (%). ^c^ 1 ≤ USD 50 K, 2 ≥ USD 50 and <10 K, 3 ≥ USD 10 K. ^d^ 1 ≤ HS Diploma; HS Diploma/GED, 2 = Some college, 3 = Bachelor, 4 = Postgraduate. ^e^ 0 = 0 days; 1 = 1 day; 2 = 2 days; 3 = 3 days; 4 = 4 days; 5 = 5 days; 6 = 6 days; 7 = 7 days.

**Table 2 nutrients-14-04643-t002:** Associations between prenatal caffeine exposure, total sugar intake, brain activation to reward anticipation, thickness measurements of taste-processing regions.

	Daily vs. No (β; 95% CI; *p*)	Weekly vs. No (β; 95% CI; *p*)	Less than weekly vs. No (β; 95% CI; *p*)
Total Sugar intake (gm)	3.5; 1.17–5.76; 0.003 ^ab^	0.63; −1.76–3.02; 0.60	0.51; −2.07–3.09; 0.69
BMI (kg/m^2^)	0.45; 0.19–0.71; <0.001 ^ab^	0.28; 0.01–0.55; 0.03 ^a^	−0.00; −0.29–0.29; 0.99
Activation in anticipation of large reward			
Rostral MFC	−0.03; −0.06–0.00; 0.02 ^a^	0.004; −0.02–0.03; 0.76	0.02; −0.01–0.04; 0.28
Caudal MFC	−0.02; −0.03–0.00; 0.06	0.002; −0.01–0.02; 0.78	−0.01; −0.02–0.01; 0.54
Medial OFC	0.01; −0.03–0.06; 0.58	0.05; 0.00–0.11; 0.04 ^c^	0.08; 0.02–0.14; 0.004
Lateral OFC	−0.00; −0.04–0.03; 0.78	0.03; −0.01–0.07; 0.12	0.04; 0.00–0.09; 0.03 ^c^
Rostral ACC	−0.02; −0.05–0.00; 0.05	0.00; −0.02–0.02; 0.94	0.003; −0.02–0.03; 0.81
Caudal ACC	−0.02; −0.03–0.00; 0.02	−0.01; −0.02–0.01; 0.52	0.004; −0.01–0.02; 0.69
Accumbens	−0.02; −0.04–0.01; 0.34	0.02; −0.00–0.06; 0.10	0.006; −0.03–0.04; 0.72
Amygdala	−0.01; −0.04–0.002; 0.34	−0.004; −0.02–0.01; 0.68	0.00; −0.02–0.02; 0.99
Insula	−0.01; −0.03–0.00; 0.05	−0.006; −0.02–0.009; 0.44	−0.003; −0.02–0.01; 0.69
Thalamus	−0.01; −0.03–0.00; 0.01 ^c^	−0.002; −0.01–0.01; 0.76	−0.002; −0.02–0.01; 0.76
Activation in anticipation of reward			
Rostral MFC	−0.02; −0.04–0.00; 0.05	0.003; −0.02–0.03; 0.78	0.009; −0.01–0.03; 0.46
Caudal MFC	−0.007; −0.02–0.00; 0.27	0.003; −0.01–0.01; 0.64	−0.002; −0.01–0.01; 0.79
Medial OFC	0.00; −0.03–0.05; 0.80	0.04; 0.002–0.09; 0.03^c^	0.06; 0.01–0.11; 0.01 ^c^
Lateral OFC	0.00; −0.02–0.03; 0.74	0.02; −0.007–0.06; 0.12	0.03; −0.00–0.07; 0.05
Rostral ACC	−0.01; −0.03–0.00; 0.24	0.006; −0.01–0.03; 0.62	0.008; −0.01–0.03; 0.50
Caudal ACC	−0.00; −0.02–0.00; 0.27	−0.00; −0.01–0.01; 0.79	0.00; −0.01–0.02; 0.42
Accumbens	−0.00; −0.03–0.02; 0.87	0.03; 0.004–0.06; 0.02 ^c^	0.01; −0.01–0.04; 0.35
Amygdala	−0.01; −0.03–0.005; 0.15	−0.00; −0.02–0.01; 0.92	0.00; −0.02–0.02; 0.95
Insula	−0.00; −0.01–0.007; 0.38	−0.00; −0.01–0.01; 0.84	0.00; −0.01–0.01; 0.84
Thalamus	−0.00; −0.02–0.004; 0.17	0.00; −0.01–0.01; 0.85	0.00; −0.01–0.01; 0.86
Thickness (mm)			
Insula	0.01; 0.00–0.01; 0.03 ^a^	0.01; 0.00–0.02; 0.03 ^a^	−0.00; −0.01–0.01; 0.81
Medial OFC	0.01; −0.00–0.01; 0.25	0.01; 0.00–0.02; 0.03	0.01; −0.00–0.01; 0.22
Lateral OFC	0.00; −0.00–0.01; 0.75	0.002; −0.00–0.01; 0.67	0.00; −0.01–0.01; 0.70
Rostral ACC	0.00; −0.01–0.01; 0.33	−0.008; −0.02–0.01; 0.23	−0.00; −0.01–0.01; 0.99
Caudal ACC	0.00; −0.02–0.02; 0.84	0.009; −0.01–0.03; 0.46	−0.01; −0.03–0.02; 0.67

Note: Here, No = No prenatal caffeine exposure; Daily = at least once a day prenatal caffeine exposure; Weekly = less than once a day but more than once a week prenatal caffeine exposure; Less than weekly = less than once a week prenatal caffeine exposure. MFC, middle frontal cortex, ACC; anterior cingulate cortex; OFC orbitofrontal cortex. ^a^ significance with *p* < 0.05 after covariates adjustments. ^b^ significance that survived Bonferroni-correction for multiple comparisons with *p* < 0.05. ^c^ indicates *p* values that appeared to be below the threshold of *p* < 0.05 on pairwise comparison however, since no main group effects for these outcomes were seen we did not consider them as a significant observation

**Table 3 nutrients-14-04643-t003:** Post hoc comparisons with Bonferroni correction for multiple comparisons.

Group Pairwise Comparisons	Marginal Mean Difference	95% CI	*p* Bonferroni-Corrected (Pairwise)
Total Sugar intake			
Daily vs. No	3.508	0.422–6.594	0.01
Within limit vs. No	3.26	0.13–6.40	0.03
Above limit vs. No	9.79	2.51–17.06	0.004
BMI			
Daily vs. No	0.448	0.097–0.799	0.004
Weekly vs. No	0.285	−0.079–0.650	0.03
Within limit vs. No	0.451	0.113–0.789	0.004
Above limit vs. No	0.931	0.149–1.713	0.01
Rostral MFC activation in anticipation of large reward			
Daily vs. less than weekly	−0.042	−0.08–9.531 × 10^−5^	0.04

Note: Here, prenatal caffeine exposure No, No exposure; Daily, at least once a day; Weekly, less than once a day but more than once a week; Less than weekly, less than once a week; Within limit, up to 2 cups of daily prenatal caffeine exposure; Above limit, 3 or more cups of daily prenatal caffeine exposure. All listed post hoc results are two sided and Bonferroni corrected *p* < 0.05. Nonsignificant comparisons are not listed.

## Data Availability

All study data are included in the article. The analysis codes will be made available on request.

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
