# Peer review of "Prenatal Caffeine Exposure Is Linked to Elevated Sugar Intake and BMI, Altered Reward Sensitivity, and Aberrant Insular Thickness in Adolescents: An ABCD Investigation"

_nutrients, 2022, doi:10.3390/nu14214643_

Round 1

Reviewer 1 Report

This manuscript describes an analysis of interrelationships among prenatal caffeine exposure (PCE) and pediatric total sugar intake (BSI), various imaging-based measures of brain structure, brain activation changes during anticipation of award, and BMI.  Strengths of this work include leveraging data from the ABCD cohort study, which is a large and well-described longitudinal study of child development that includes an exceptional amount of brain imaging data.  Taken together, these data should be sufficient to support a rigorous, well-powered analysis of these relationships and contribute meaningfully to the literature, addressing current knowledge gaps as to how PCE may increase risk of childhood obesity.  However, there are shortcomings in the approach, the interpretation of the results, and the manuscript itself that need major improvements before this work is publishable.

For example, some major concerns:

1.     Abstract - The abstract should be rewritten in a more traditional style for a quantitative analysis of this type:  clear presentation of the research questions (there are multiple questions addressed in this study), brief description of the study population and the analytic approach, summary of all findings (not just ones that support the hypothesis) and a discussion/conclusion statement.  The current abstract is very muddled and does not present key information needed to understand what was done and what the findings were.  The first sentence gives some background and indicates uncertainty as to whether relationships between PCE and obesity are mediated by sugar preference, but there are no clear research questions stated.  The only information about the statistical approach is the phrase “using linear mixed models” which is not at all adequate for the number and complexity of different statistical models conducted in this study.  The note about participant exclusions is important but can be reserved for the main text – currently it takes up a lot of the allowed word count.  A major statement in the abstract (“We show for the first time…”) is not supported by the analysis conducted and the findings.  Only one set of numerical results is presented (variation in relationship between TSI and BMI by child sex/gender), though this presumable a minor aspect of the study findings because it is not discussed at all in the Discussion.

2.     Description of methods – Section 2.1 should include a clear and succinct description of the ABCD study with a link to key publications with detailed methods.  There should be mention of the longitudinal nature of ABCD and the timepoints at which data for this analysis were collected. Most of the data used was presumably collected at the 9-11 year visit (though, unclear if this is 1 or 2 visits as the Discussion (line 338) mentions that data were collected on 9 and 11 year children, which implies 2 visits), but some likely were not, including the caffeine survey and some covariates.  Note that the use of the word “baseline” here and in other parts of the manuscript is probably incorrect; in the context of a longitudinal study, baseline describes data collected at the earliest timepoint (e.g., enrollment visit).  Respondents to each survey should be clearly listed (e.g., did kids complete the dietary survey, or caregivers?).  Statistical models need to be more carefully described, including the approach to selection of covariates (possible confounders).  The grouping and order of statistical models presented is confusing (e.g., lines 138-141).  Why focus on relationships between PCE and TSI first, and then investigate PCE and other outcomes (BMI, brain activity and brain structure) secondarily, as a group?  The high-level conceptual approach is not explained.  If the goal is to determine whether TSI, brain activity and brain structure mediate the previously established relationships between PCE and BMI, it might make more sense to first examine the main effects (PCE-BMI), then the relationship between PCE and each mediator, each mediator and BMI, and then explore an advanced mediation model for possible mediating variables which show evidence of playing this role in the preceding models.  If the goal is to determine whether the relationships between PCE and TSI are mediated by brain activity and structure, a different grouping/ordering may be appropriate.  I’m not sure what the guiding research questions and objectives are. Also, the purpose of examining effect modification by child sex/gender is not at all explained.  Lines 151-154 are unclear – they imply that the second sentence (post-hoc analyses) were performed for the relationships of covariates described in first sentence (relationships with what?  PCE, based on Table 1), but I don’t think that’s the case.

3.     Presentation of results – I recommend that the findings be structured by sections guided by the research questions and the conceptual model.  Subsections would be useful for this.  See minor notes about presentation of results below as well.

4.     Interpretation and discussion of results – The paper would be improved by a more organized and thorough discussion of findings.  For example, it would help if the first paragraph summarized the most important findings.  Rather, most of this paragraph is devoted to a secondary analysis involving soft drinks that was barely mentioned in the Methods and Results and not mentioned at all in the Abstract.  This is a distraction.  Other main findings highlighted in the Results section are somewhat buried in the Discussion.  The mediation analysis is not mentioned at all, and the differences in associations between boys and girls is noted but not interpreted or contextualized. It’s still unclear if this is an important aspect of the analysis or just an afterthought/side-analysis.  Finally, the limitations section should be expanded and more thoughtful.  Small effect sizes are not a methodological limitation (in fact, detection of small associations are only possible w/ large sample sizes, a strength), but rather a challenge in extrapolating the findings to public health relevance.  Several other limitations of this study should be presented here, instead.

Minor concerns

The manuscript would benefit from additional editing.  Some of the language is too casual for an academic journal.  The tables and figures are confusing in some aspects.  For example:

Figure 2 – the ordering of PCE groups on the x-axis do not go in order of exposure magnitude.

All figures – the captions and figures should contain sufficient information to stand alone, for the reader to interpret results.  Captions should briefly describe statistical methods used to generate the results displayed (including all adjustment variables, rather than referring back to the main text), and should not include any summary or interpretation of the findings, which is done in the Results and Discussion sections. 

Table 2 – the note is fairly confusing and should be rewritten.  It’s unclear if these are crude or adjusted associations.  One number has an asterisk (*) but this is not defined in the table note or in the main text, unless I missed it.  The note says that footnote “a” flags results significant at a threshold of 0.05, but there are several p-values <0.05 that are not flagged.  Why not?

Author Response

Q1: Abstract - The abstract should be rewritten in a more traditional style for a quantitative analysis of this type:  clear presentation of the research questions (there are multiple questions addressed in this study), brief description of the study population and the analytic approach, summary of all findings (not just ones that support the hypothesis) and a discussion/conclusion statement.  The current abstract is very muddled and does not present key information needed to understand what was done and what the findings were.  The first sentence gives some background and indicates uncertainty as to whether relationships between PCE and obesity are mediated by sugar preference, but there are no clear research questions stated.  The only information about the statistical approach is the phrase “using linear mixed models” which is not at all adequate for the number and complexity of different statistical models conducted in this study.  The note about participant exclusions is important but can be reserved for the main text – currently it takes up a lot of the allowed word count.  A major statement in the abstract (“We show for the first time…”) is not supported by the analysis conducted and the findings.  Only one set of numerical results is presented (variation in relationship between TSI and BMI by child sex/gender), though this presumable a minor aspect of the study findings because it is not discussed at all in the Discussion.

Answer: Thank you for the above comments and we apologize for the lack of clarity.   

As suggested, we have edited the abstract for better clarity. These changes have been made in the abstract section of the revised manuscript. Please see page no. 2 in the revised manuscript.

Q2: Description of methods – Section 2.1 should include a clear and succinct description of the ABCD study with a link to key publications with detailed methods.  There should be mention of the longitudinal nature of ABCD and the timepoints at which data for this analysis were collected. Most of the data used was presumably collected at the 9–11-year visit (though, unclear if this is 1 or 2 visits as the Discussion (line 338) mentions that data were collected on 9- and 11-year children, which implies 2 visits), but some likely were not, including the caffeine survey and some covariates.  Note that the use of the word “baseline” here and in other parts of the manuscript is probably incorrect; in the context of a longitudinal study, baseline describes data collected at the earliest timepoint (e.g., enrolment visit).  Respondents to each survey should be clearly listed (e.g., did kids complete the dietary survey, or caregivers?). 

Answer: Thank you for your valuable comment.  

We appreciate this point, and we have added a section on the ABCD study description including details on the longitudinal nature of data collection in the methods section of the revised manuscript. The word “baseline” has been replaced by “enrolment” for clarity. Also, the clarification on the respondents to each survey has been added in the methods section of the revised manuscript.

Please see page nos. 4-8 of the revised manuscript.

Q3: Statistical models need to be more carefully described, including the approach to selection of covariates (possible confounders). The grouping and order of statistical models presented is confusing (e.g., lines 138-141). Why focus on relationships between PCE and TSI first, and then investigate PCE and other outcomes (BMI, brain activity and brain structure) secondarily, as a group? The high-level conceptual approach is not explained. If the goal is to determine whether TSI, brain activity and brain structure mediate the previously established relationships between PCE and BMI, it might make more sense to first examine the main effects (PCE-BMI), then the relationship between PCE and each mediator, each mediator and BMI, and then explore an advanced mediation model for possible mediating variables which show evidence of playing this role in the preceding models. If the goal is to determine whether the relationships between PCE and TSI are mediated by brain activity and structure, a different grouping/ordering may be appropriate.  I’m not sure what the guiding research questions and objectives are. Also, the purpose of examining effect modification by child sex/gender is not at all explained. Lines 151-154 are unclear – they imply that the second sentence (post-hoc analyses) were performed for the relationships of covariates described in first sentence (relationships with what?  PCE, based on Table 1), but I don’t think that’s the case.

Answer: Thank you for this great suggestion. 

We have edited the methods and the results sections in the revised manuscript to convey the statistical approach and the results with better clarity and in the order suggested by the reviewer in the revised manuscript. Please see page nos. 8-9 of the revised manuscript

Q4: Presentation of results – I recommend that the findings be structured by sections guided by the research questions and the conceptual model.  Subsections would be useful for this.  See minor notes about presentation of results below as well.

Answer: Thank you for this suggestion. We have presented the results in different sections as per the reviewer’s suggestions. Please see the results section (page nos. 11-13 in the revised manuscript.

Q5: Interpretation and discussion of results – The paper would be improved by a more organized and thorough discussion of findings.  For example, it would help if the first paragraph summarized the most important findings.  Rather, most of this paragraph is devoted to a secondary analysis involving soft drinks that was barely mentioned in the Methods and Results and not mentioned at all in the Abstract.  This is a distraction.  Other main findings highlighted in the Results section are somewhat buried in the Discussion.  The mediation analysis is not mentioned at all, and the differences in associations between boys and girls is noted but not interpreted or contextualized. It’s still unclear if this is an important aspect of the analysis or just an afterthought/side-analysis.  Finally, the limitations section should be expanded and more thoughtful.  Small effect sizes are not a methodological limitation (in fact, detection of small associations are only possible w/ large sample sizes, a strength), but rather a challenge in extrapolating the findings to public health relevance.  Several other limitations of this study should be presented here, instead.

Answer: Thank you for this suggestion. We have organized the discussion of the results in the revised manuscript. The “Strengths and limitations” section has been rearranged and elaborated as per the suggestions of the reviewer.  Please see page nos. 16-21 in the revised manuscript.

Q6: Figure 2 – the ordering of PCE groups on the x-axis do not go in order of exposure magnitude.

Answer: Thank you for this suggestion. We have made the edits in Figure 2 as suggested.

Q7: All figures – the captions and figures should contain sufficient information to stand alone, for the reader to interpret results.  Captions should briefly describe statistical methods used to generate the results displayed (including all adjustment variables, rather than referring to the main text), and should not include any summary or interpretation of the findings, which is done in the Results and Discussion sections. 

Answer: Thank you for pointing this. We have restructured the figure and rephrased the captions as suggested in the revised version of the manuscript.

Q8: Table 2 – the note is fairly confusing and should be rewritten.  It’s unclear if these are crude or adjusted associations.  One number has an asterisk (*) but this is not defined in the table note or in the main text, unless I missed it.  The note says that footnote “a” flags results significant at a threshold of 0.05, but there are several p-values <0.05 that are not flagged.  Why not?

Answer: Thank you for pointing this. We have rephrased the note for Table 2 for better clarity. Please see page no. in the revised manuscript.

Reviewer 2 Report

The authors present findings important to public health. Prior research evidence is unclear on how to best counsel pregnant women regarding caffeine intake.

The findings are based on a large cohort study and the manuscript is well written. I offer a few comments:

1) Why is BMI-for-age percentage not used? This is a preferred measure of BMI in children/adolescents.

2) Table 2: Indicate the meaning of *

Author Response

Q1: Why is BMI-for-age percentage not used? This is a preferred measure of BMI in children/adolescents.

Answer: We thank the reviewer for critically evaluating the manuscript and we apologize for the lack of clarity. We have now edited the BMI calculation method for clarity in the revised manuscript.

The following method was adopted for BMI calculation as per CDC guidelines (https://www.cdc.gov/nccdphp/dnpao/growthcharts/training/bmiage/page5_2.html):

703 × weight(lbs)/height(in)2 .

Please see page no. 7 of the revised manuscript for edits.

 Q2: Table 2: Indicate the meaning of *

Answer: Thank you for pointing this we have made the edits in Table 2 legend in the revised manuscript for clarity.

Round 2

Reviewer 1 Report

There are some improvements in the manuscript with this round of revisions, but I feel that it still needs substantial editing before being fit for publication.

The writing is unclear in several places.  For example, the first sentence of Section 2.8 does not indicate what comparisons are used for the statistical tests listed.  There are incomplete sentences in the manuscript and grammar errors.  Many sections seem to have been written in a rush.

It's unclear why the analysis of differences by sex/gender were removed.  The new draft reads, "Furthermore, the study is limited by the assessment of study findings based on sex and ethnicity differences as this was beyond the scope of the current investigation."  This is very confusing to me, as the sentence implies that these analyses were conducted, yet also that a limitation of the work was that it was "beyond the scope" of the project.  Why would it be beyond the scope of the study, especially given that sex differences were explored in the first version of the manuscript?

Author Response

Manuscript ID: 1918030

Title: " Prenatal Caffeine Exposure is Linked to Elevated Sugar Intake and BMI, Altered Reward Sensitivity, and Aberrant Insular Thickness in Adolescents: an ABCD Investigation "

Response to the Referees Report

We gratefully acknowledge the reviewer’s suggestions and critical feedback, which certainly improved the significance and quality of our manuscript. In the following, we answered the questions, and included the same in the revised version of our manuscript (the changes are highlighted).

We hope that we have addressed the concerns of the reviewers.

Profound thanks in advance.

Sincerely,

Paule V. Joseph Ph.D., MS, FNP-BC, CTN-B, FAAN

Lasker Clinical Research Scholar

Tenure-Track Investigator

Comments of Reviewer # 1 and Responses

Q1: The writing is unclear in several places.  For example, the first sentence of Section 2.8 does not indicate what comparisons are used for the statistical tests listed. 

Answer: Thank you for the above comment and we apologize for the lack of clarity.   We have edited the section for clarity.

Q2: There are incomplete sentences in the manuscript and grammar errors.  Many sections seem to have been written in a rush.

Answer: Thank you for your valuable comment.  We have worked on this comment and have edited the manuscript throughout.

Q3: It's unclear why the analysis of differences by sex/gender were removed.  The new draft reads, "Furthermore, the study is limited by the assessment of study findings based on sex and ethnicity differences as this was beyond the scope of the current investigation."  This is very confusing to me, as the sentence implies that these analyses were conducted, yet also that a limitation of the work was that it was "beyond the scope" of the project.  Why would it be beyond the scope of the study, especially given that sex differences were explored in the first version of the manuscript?

Answer: We have now added a separate section for sex differences in the revised version of the manuscript as suggested.
